# NIgPred: Class-Specific Antibody Prediction for Linear B-Cell Epitopes Based on Heterogeneous Features and Machine-Learning Approaches

**DOI:** 10.3390/v13081531

**Published:** 2021-08-03

**Authors:** Chi-Hua Tung, Yi-Sheng Chang, Kai-Po Chang, Yen-Wei Chu

**Affiliations:** 1Department of Bioinformatics, Chung Hua University, Hsinchu City 300, Taiwan; chihua.tung@chu.edu.tw; 2Department of Optoelectronics and Materials Engineering, Chung Hua University, Hsinchu City 300, Taiwan; 3Institute of Genomics and Bioinformatics, National Chung Hsing University, 145 Xingda Rd., South Dist., Taichung City 402, Taiwan; 103025048@gm.asia.edu.tw; 4Ph.D. Program in Medical Biotechnology, National Chung Hsing University, 145 Xingda Rd., South Dist., Taichung City 402, Taiwan; d17179@mail.cmuh.org.tw; 5Department of Pathology, China Medical University Hospital, Taichung City 404, Taiwan; 6Institute of Molecular Biology, National Chung Hsing University, 145 Xingda Rd., South Dist., Taichung City 402, Taiwan; 7Agricultural Biotechnology Center, National Chung Hsing University, 145 Xingda Rd., South Dist., Taichung City 402, Taiwan; 8Biotechnology Center, National Chung Hsing University, 145 Xingda Rd., South Dist., Taichung City 402, Taiwan; 9Ph.D. Program in Translational Medicine, National Chung Hsing University, 145 Xingda Rd., South Dist., Taichung City 402, Taiwan; 10Rong Hsing Research Center for Translational Medicine, National Chung Hsing University, 145 Xingda Rd., South Dist., Taichung City 402, Taiwan

**Keywords:** B-cell epitope, machine learning, feature selection, antigen, antibody

## Abstract

Upon invasion by foreign pathogens, specific antibodies can identify specific foreign antigens and disable them. As a result of this ability, antibodies can help with vaccine production and food allergen detection in patients. Many studies have focused on predicting linear B-cell epitopes, but only two prediction tools are currently available to predict the sub-type of an epitope. NIgPred was developed as a prediction tool for IgA, IgE, and IgG. NIgPred integrates various heterologous features with machine-learning approaches. Differently from previous studies, our study considered peptide-characteristic correlation and autocorrelation features. Sixty kinds of classifier were applied to construct the best prediction model. Furthermore, the genetic algorithm and hill-climbing algorithm were used to select the most suitable features for improving the accuracy and reducing the time complexity of the training model. NIgPred was found to be superior to the currently available tools for predicting IgE epitopes and IgG epitopes on independent test sets. Moreover, NIgPred achieved a prediction accuracy of 100% for the IgG epitopes of a coronavirus data set. NIgPred is publicly available at our website.

## 1. Introduction

There are many pathogens, such as bacteria, viruses, and allergens, which are present in daily life. When foreign objects invade the human body, the immune system attacks them. The human immune system has natural immunity and acquired immunity against invading antigens. Innate immunity, supported by dendritic cells and macrophages, is the first line of defense and is not specific to the invading antigens. Compared with innate immunity, acquired immunity has specificity and diversity, and involves B-cells and T-cells. B-cells are responsible for secreting antibodies to fight bacteria, while T-cells are responsible for defending against viruses and fungi. Antibodies bind to specific protein sequences on antigens, and the binding positions of these antigens and antibodies are called epitopes [1].

When foreign antigens enter the human body, they can recognize foreign antigens or neutralize similar pathogens and, thus, can help with vaccine development [2]. An antibody is composed of two light chains and two heavy chains; heavy chains can be classified into five categories, namely, IgG, IgA, IgE, IgM, and IgD. IgG is the most numerous epitope, which is responsible for fighting foreign viruses and bacteria. IgA, meanwhile, can protect the digestive tract against viruses. If IgG loses its ability to function, IgM steps in to eliminate germs. IgA and IgG can also be used as early screening markers for lung cancer [3]. In a similar vein, over the past couple years, severe acute respiratory syndrome coronavirus 2 (SARS-CoV-2) has affected tens of millions of people worldwide, and precise and effective identification of asymptomatic infections is needed urgently. Quan-Xin Long et al. [4] found IgG antibodies in all assessed patients 19 days after being infected with the virus. As for the remaining two categories of heavy chains, IgD has the fewest epitopes, and it can help IgE to perform basophil separation [5]. IgE is closely related to food allergies [6]. In the United States, approximately 200 people die from food allergies every year [7]; BCIgEPred [8] is a tool specifically aimed at improving the accuracy of predicting IgE.

B-cell epitopes can be divided into two types; namely, linear epitopes and conformational epitopes (also called discontinuous epitopes in some studies). In nature, approximately 90% of epitopes are conformational [9], which are harder to predict and require more structural information than linear epitopes; for example, the ROC curve of the epitope predictor tool BepiPred [10] only achieved 57% accuracy in a case study. Therefore, linear epitopes are used by scientists to quickly understand the relationships between epitopes and diseases. In the past 20 years, many tools, such as BcePred [11], BCPred [12], SVMTrip [13], APCpred [14], Lbtope [15], iBCE-EL [16], and DLBEpitope [17], have been developed to identify whether a given sequence is a linear epitope. Yet, these tools only detect whether the epitope is a linear epitope; they cannot predict whether the epitope is IgG, IgE, or IgA. Only some tools, such as BCIgEPred, AlgPred [18], and IgPred [19], can predict the sub-type of an epitope. The authors who developed BCIgEPred collected 1273 positive data sets and 4226 negative data sets to evaluate the model. Unlike in previous studies, they considered using the exact epitopes as the training set to improve model learning. As the training data set classes were imbalanced, they used random sampling to divide the data categories into one-to-one relationships, in order to build a balanced training data set. In addition, they used a composition of weighted diamino acids for encoding purposes and applied the random forest and support vector machine (SVM) algorithms as classifiers. After training, the balance accuracy and the Matthews correlation coefficient (MCC) reached 76% and 51%, respectively, on the independent IgE test set.

AlgPred [18] is a tool developed by the Bioinformatics Center in India to predict allergenic proteins. This tool integrates four different methods and uses the SVM classifier to predict antiallergens, with an accuracy of up to 85% and an MCC of 70%. IgPred was the first tool developed to predict specific B-cell epitopes; six different feature-encoding methods and eight different Waikato Environment for Knowledge Analysis (WEKA) [20] algorithms were used to build the model. As the lengths of epitopes are different, the authors compared the performance of fixed- and variable-length epitopes. The accuracy of the fixed-length epitope reached 72%, and the MCC was 46% for the independent data set. For the independent test set of the variable-length epitope, the accuracy and MCC were 74% and 49%, respectively.

At present, existing linear antigen prediction tools all have epitope length constraints. As the amount of B-cell data continues to increase, it is very important to improve the prediction length limitation. We designed the NIgPred tool to increase the maximum length of a predictable epitope and improve the accuracy of predicting IgE, IgG, and IgA. Most previous studies have only used the compositions and physico-chemical properties of amino acids and could not fully express the mutual influence of each amino acid. Therefore, we added two major types of features, namely autocorrelation features and peptide-characteristic correlation features, to increase the interactions between the calculated amino acids. At the same time, to reduce the complexity of the model and improve its accuracy, we used a genetic algorithm and a hill-climbing algorithm to select features and execute 60 different WEKA algorithms, respectively. Finally, to verify the accuracy of the NIgPred model, we collected IgE and SARS-CoV-2 IgG case studies, and reached accuracies of approximately 0.79 and 1, respectively. Furthermore, the accuracy of our model for the case study of IgE and IgG (Section 3.6) was superior to that of the existing tools (i.e., BCIgEPred and IgPred).

## 2. Materials and Methods

### 2.1. Overview of Our Study

Figure 1 shows the main flowchart of the process presented in this study. The data collected from the Immune Epitope Database (IEDB [21]; see the “Data set and Pre-processing” section) were processed and divided into four groups: IgA epitope, IgG epitope, IgE epitope, and non-B-cell epitopes (non-BCE). The four-part data set was run through CD-HIT, to remove similar sequences, and was then divided based on a cutoff value of 0.7. After CD-HIT discarded the duplicate data, to verify the model performance, we randomly split the IgA, IgE, and IgG data sets, with 80% for training and 20% for testing, and these data sets were further separated into positive and negative data sets containing B-cell epitopes or non-B-cell epitopes. For instance, the IgA data set was composed of positive IgA data and negative IgE, IgG, and non-BCE data.

Unbalanced data sets can lead to biases in machine learning algorithms; thus, an appropriate positive/negative data set ratio (P/N ratio) is necessary before constructing a prediction model. To resolve these issues, we adopted the NearMiss sampling method for the negative data sets. Then, the P/N ratio was 1:1 for the IgA, IgE, and IgG models.

Feature encoding is an important factor in determining inferences from the training model. Therefore, we utilized two major categories of feature encoding: Sequence-based and physico-chemical-based encoding. Sequence-based encoding can be divided based on composition features and peptide-characteristic correlation features, and physico-chemical-based encoding can be separated based on autocorrelation features and physico-chemical features. Finally, 1132 dimensions were obtained for the feature space.

Feature selection was then conducted, to determine whether the model could be optimized. Feature selection and optimization were followed by a comparison between the results of the genetic algorithm, hill-climbing algorithm, and 60 different WEKA algorithms, in order to select the optimal algorithm.

To prove the rationality of the NIgPred system, we compared the prediction results of the four separate types of features in the model. Furthermore, to examine the impact of epitope features on prediction accuracy and determine their importance for the training system, we used a feature selection method to determine important features (Appendix A).

### 2.2. Data Set and Pre-Processing

We collected epitope data from the IEDB, screened out the data with linear epitope information, and designated the positive B-cell data set as BCES (B-cell epitopes). The negative B-cell data set was denoted as non-BCES. To avoid excessive data redundancy, we used the CD-HIT [22] tool to remove duplicate proteins. The CD-HIT threshold was set to 0.7, and epitopes with lengths of less than five were removed. We sorted out a total of 49,093 IgG, 2435 IgE, 631 IgA, and 127,010 non-BCE samples. As the data were excessively imbalanced, we used the NearMiss function [23] in the imbalanced-learn [24] tool to balance the data. In the IgE case study, we collected 19 and 410 IgE samples from the Allergen Database for Food Safety (ADFS) [25] and the Allerbase database [26], respectively. In the IgG case study, we collected 40 related SARS-CoV-2 antibody data sets [27].

### 2.3. Feature Construction

Physico-chemical properties are commonly used in the study of biological information. In previous studies using IgPred, it has been found that diamino acid composition was the most effective feature. In this study, we used four major types of features for encoding, namely composition, physico-chemical, autocorrelation, and peptide-characteristic correlation features. We applied the Peptides [28] and Protr [29] packages in the R programming language to encode protein sequences, which were compiled into 1132 feature codes. In terms of composition characteristics, we adopted the monoamino acid composition, epitope protein sequence length, diamino acid composition, and composition transition distribution (CTD). As the composition and transition components of the CTD are similar to those of other features, we did not use them.

The physico-chemical properties were calculated using the Peptides package (Table 1). For autocorrelation, we used the Moreau–Broto [30], Moran [31], and Geary [32] approaches, and the peptide-characteristic correlation features were obtained by applying the Protr package for the calculations. See Appendix A for explanations of the specific package functions.

### 2.4. Machine Learning and Feature Selection Algorithms

To select the best algorithm and reduce the complexity of the model, we executed 60 kinds of WEKA algorithms and two kinds of feature selection algorithms (genetic and hill-climbing algorithms). The genetic algorithm (GA) [33] first randomly generates an initial population, calculates the fitness function of each chromosome, orders the fitness values, and then selects the chromosome with the greater fitness to be copied. After replication, it crosses over the chromosomes and mutates some of them. Using mutations can enable the algorithm to avoid falling into local optima. The hill-climbing algorithm (HC) [34] randomly selects an initial point and compares it with the surrounding points. If the selected point is the largest, it is used as the node until reaching the highest point. However, the hill-climbing algorithm can easily fall into local optima. Hence, in this study, we used the hill-climbing and genetic algorithms together to select features. Finally, after comparing different algorithms and feature selection methods, we adopted a sequential minimal optimization (SMO) [35] algorithm and the hill-climbing algorithm for the IgA model, the FLDA [36] algorithm and genetic algorithm for the IgE model, and the random forest [37] algorithm and hill-climbing algorithm for the IgG model.

### 2.5. Validation Measures

The performance of the classifier was evaluated and compared, based on the accuracy of its prediction. In this study, a five-fold cross-validation model was used to verify the training model. The data were divided into five parts, of which four were used as training data, while the remaining part was used as test data. The remaining data were tested sequentially. For the problem of binary classification, the prediction results may be of four different types: True positives (TPs), false positives (FPs), false negatives (FNs), and true negatives (TNs). We aimed to predict protein epitope sequences. Finally, we adopted the Matthews correlation coefficient (MCC), accuracy (ACC), precision, and recall metrics to evaluate the model. These metrics are defined in the following formulas:$$\mathrm{MCC}=\frac{\left(\mathrm{TP}\times \mathrm{TN}\right)-\left(\mathrm{FN}\times \mathrm{FP}\right)}{\sqrt{\left(\mathrm{TP}\;+\;\mathrm{FN}\right)\times \left(\mathrm{TN}\;+\;\mathrm{FP}\right)\times \left(\mathrm{TP}\;+\;\mathrm{FP}\right)\times \left(\mathrm{TN}\;+\;\mathrm{FN}\right)}},$$
$$\mathrm{Precision}=\frac{\mathrm{TP}}{\mathrm{TP}\;+\;\mathrm{FP}},$$
$$\mathrm{ACC}=\frac{\mathrm{TP}\;+\;\mathrm{TN}}{\mathrm{TP}\;+\;\mathrm{FP}\;+\;\mathrm{TN}\;+\;\mathrm{FN}},$$
$$\mathrm{Recall}=\frac{\mathrm{TP}}{\mathrm{TP}\;+\;\mathrm{FN}}.$$

## 3. Results and Conclusions

### 3.1. Amino Acid Composition

To understand the data distribution of the epitopes, we used the monoamino acid composition feature to analyze the four aforementioned data sets: IgA epitope, IgE epitope, IgG epitope, and non-BCE (Figure 2). We determined the differences in the average amino acid propensities of the IgA epitope data set, IgE epitope data set, IgG epitope data set, and non-BCE data sets. Compared with other epitope isotypes, the IgA epitopes contained more Pro and Phe residues. The IgE epitopes, meanwhile, contained more Cys, Tyr, and Ile residues than other epitopes. Finally, the IgG epitopes contained more His, Met, and Trp residues than other isotypes. These results are similar to those of a previous study on IgPred [19].

### 3.2. Performance Comparisons for Various Features

In this study, we used the four major categories of features, namely composition, physico-chemical, autocorrelation, and peptide-characteristic correlation features. By using more types of features, instead of only using composition and physico-chemical features as in previous studies (Table 2), we verified the effectiveness of adding autocorrelation and peptide-characteristic correlation features. We expected that different types of features would have different effects on the training of the three isotype models, so we compared the prediction performances after model training, when the four different features were used alone and when all features were used simultaneously. As can be seen in Table 3, the results of the three models using all the features were the best, with the MCC reaching 70.7, 67.9, and 91.1 for IgA epitope, IgE epitope, and IgG epitope, respectively.

In previous studies, training and prediction using a single mixed model may result in overfitting [38]. Therefore, we concluded that there would be different characteristic distributions of epitopes corresponding to different isotypes, and that models should be established for different isotypes. We used the feature selection algorithm to rank the feature importance of each model. Significant differences in important features were found between the three models. For the IgA, IgE, and IgG models, the top 20 important features are highly diverse, as can be seen in Appendix A. In the IgA model, the composition properties were the most important features, followed by physico-chemical properties and peptide-characteristic correlation. In the IgE model, more attention was paid to composition and physico-chemical properties, and the detailed feature functions of composition properties were completely different from those of the IgA model. In the IgG model, the most significant feature was the length of the epitope, which was the same as in the IgA model, though most others used peptide-characteristic correlation.

### 3.3. Performance Comparison Based on Epitopes of Different Lengths

In this study, we collected sequences from antigen epitopes that were either long or short, with different lengths affecting the training process of the system and the predictions of the model. This study determined the lengths of the training data sets for non-BCE, IgA, IgE, and IgG. It was found that specific epitopes typically appear at specific lengths; non-BCE data were mostly of length 13–15 base pairs (Figure 3a), IgA epitope data were mostly of length 5–12 base pairs (Figure 3b), IgE epitope data were mostly of length 12–20 base pairs (Figure 3c), and IgG epitope data were mostly of length 5–12 base pairs (Figure 3d). Therefore, we chose high-frequency intervals for prediction. For the IgA model, we took epitopes of lengths 7–16 from the test set for prediction. Upon analyzing these points, the precision of the model reached 0.829, while its recall reached 0.825, ACC reached 0.825, and MCC reached 0.654. For the IgE model, we took epitopes of length 7 to 15 from the test set, in order to make predictions. Using this model, the precision reached 0.793, recall reached 0.793, ACC reached 0.791, and MCC reached 0.584. For the IgG model, epitopes with lengths from 11 to 15 were taken from the test set for prediction, and the precision, recall, ACC, and MCC reached 0.877, 0.871, 0.871, and 0.748, respectively.

### 3.4. Ability of the Model on an Independent Data Set

To verify the accuracy of the model, we randomly split the data into 80% for training and 20% for testing. At the same time, we performed five-fold cross-validation on the training set data (Table 4). The MCC values for the training data set used by the IgA, IgE, and IgG models reached 0.707, 0.679, and 0.911, respectively, and the MCC values for the test data set used by the IgA, IgE, and IgG models reached 0.715, 0.640, and 0.745, respectively. At the same time, the IgE test set was used to compare our models with the latest tool (BCIgEPred). The MCC of BCIgEPred on the test set was only 0.187.

### 3.5. Performances of the Models Trained on Data from Prior Years with Regard to Predicting Specific Epitopes Collected in Later Years

Experts in the field of antibody research care mostly about using past data to discuss recent developments. Therefore, we collected more than 30 years of data from the IEDB database to verify whether our model could represent the current spread of disease antibodies, and ordered the data by year, with the first 80% of the data set used as the training data and the remaining 20% of the later years as the test data. However, for the IgA models, the MCC results were not outstanding (Table 5). We assume that the model performance was not great due to the different data origins of the species in the IgA data set. After statistical analyses, the results show that most of the data in the early years were related to bacteria and, in recent years, the data were mostly related to viruses.

### 3.6. Case Studies

#### 3.6.1. IgE Epitope Case Study

Due to the high correlation between IgE epitopes and allergens, the current method for clinically detecting food allergies is to detect IgE. We collected data from the Allergen Database for Food Safety (ADFS) and the Allerbase database for independent verification in this study. We removed the samples from the ADFS and Allerbase databases that appeared in the IgE model training set, and obtained 19 and 410 IgE epitope samples, respectively. Then, we compared the samples with the existing tools, BCIgEPred and Igpred. After the comparison was completed, the results showed that this study achieved higher accuracy rates for these two databases than the latest BCIgEPred and IgPred tools (Table 6). The accuracies of our model for the ADFS and Allerbase databases reached 0.84 and 0.74, respectively.

#### 3.6.2. IgG Epitope Case Study

SARS-CoV-2 has affected tens of millions of people worldwide, and precise and effective identification of asymptomatic infections is needed urgently. Quan-Xin Long et al. tried to study the time between being infected by the virus and obtaining immunity and found that all patients had IgG antibodies 19 days after infection. We collected 40 IgG epitopes related to SARS-CoV-2 (Figure 4) from the Antibody Registry database, in order to verify whether our IgG model can correctly predict epitopes. These epitopes can be divided into five groups (spike protein, membrane protein, nucleocapsid protein, nucleoprotein and envelope protein). Based on the NIgPred test results, the accuracy of the IgG model reached 100% (Table 7), while the accuracy achieved by the IgPred tool for the SARS-CoV-2 data set was only 0.55.

### 3.7. Developing the NIgPred Tool

To facilitate the ease of operation for users, we developed a standalone version and web server for the NIgPred tool, in order to provide linear B-cell-specific epitope-prediction services. Users can upload epitope sequences as a FASTA-format file. Then, a prediction model (IgA, IgE, IgG) can be chosen to predict linear B-cell-specific epitopes. The website for the NIgPred tool is now available. The program interface is shown in Figure 5.

## 4. Conclusions

Antigen–antibody binding is an important part of the human immune response. This immune response helps to eliminate these pathogens when they invade our bodies. Compared with linear epitopes, conformational epitopes require much more experimental verification and, so, the prediction of linear epitopes is important. Although several prediction models for linear epitopes already exist, these linear prediction tools all have epitope length limitations. Therefore, this study was mainly conducted to improve the prediction of epitope lengths and to integrate features that were not included in previous studies. Compared with previous research, our study added autocorrelation and peptide characteristic correlation features, and used a genetic algorithm and a hill-climbing algorithm for feature selection purposes. Independent data set results confirmed that, after feature addition and feature selection were performed, the accuracy of the IgE model was improved by 0.32 and 0.49, respectively, rising above the accuracy of the latest BCIgEPred tool on the ADFS and Allerbase databases. Furthermore, the ACC of the NIgPred IgG model on a SARS-CoV-2 data set reached 1.00, with the NIgPred tool offering better performance than other tools. Finally, we set up a user-friendly website that allows users to instantly predict whether an amino acid sequence is an antigenic determinant.

## Figures and Tables

**Figure 1 viruses-13-01531-f001:**
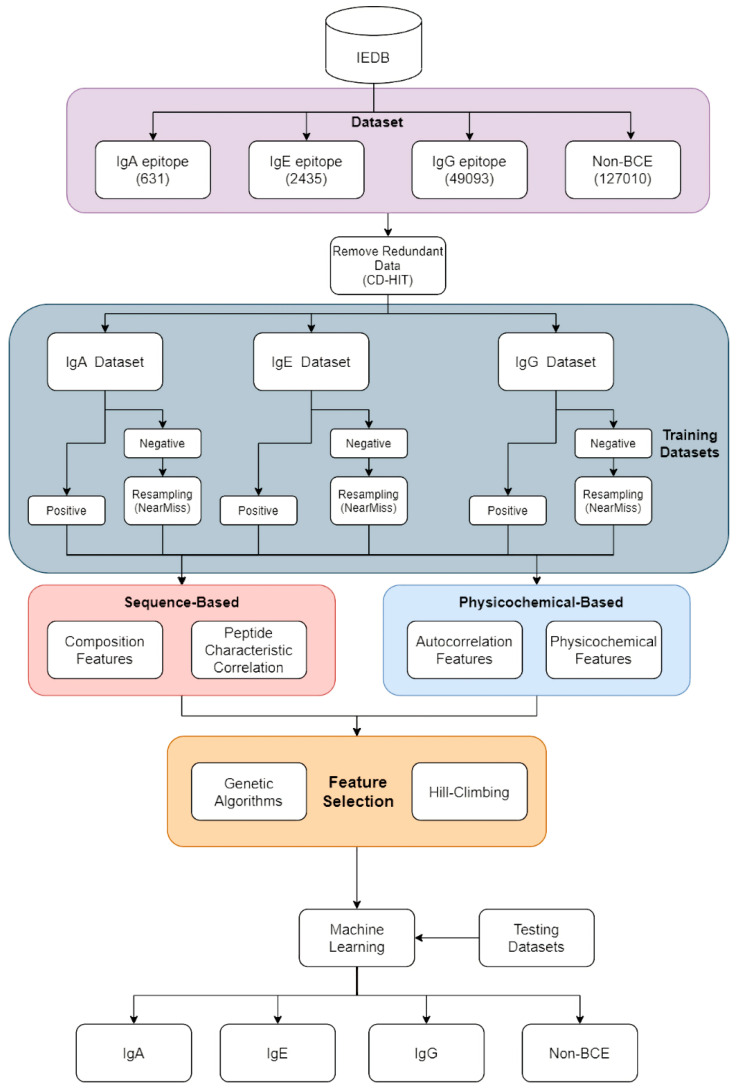
The experimental architecture of NIgPred.

**Figure 2 viruses-13-01531-f002:**
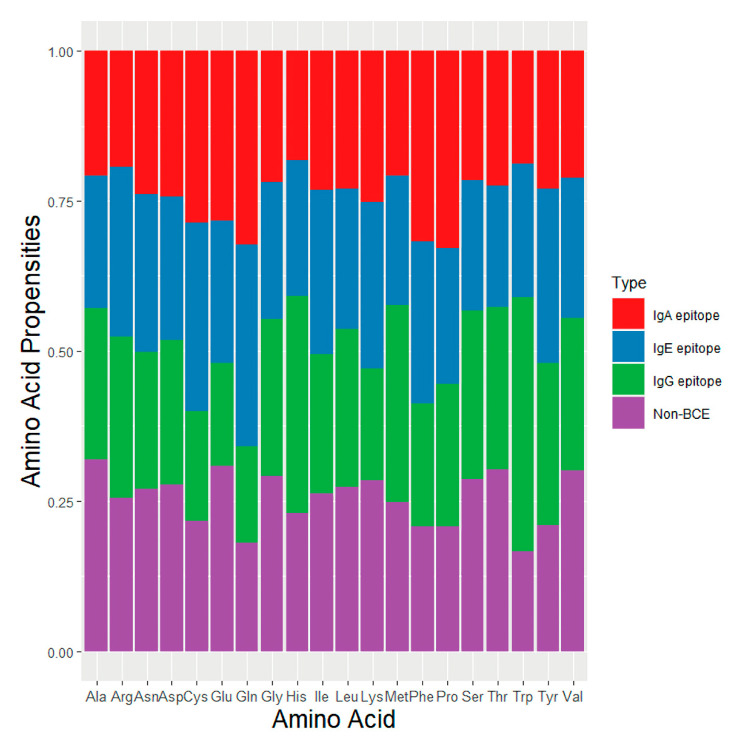
NIgPred data set of amino acid propensities.

**Figure 3 viruses-13-01531-f003:**
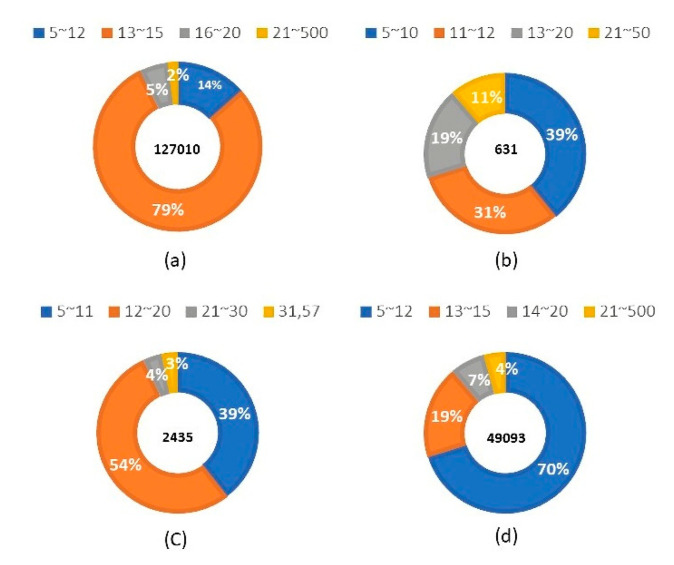
Frequencies of various epitope lengths from the following data sets: (**a**) non-BCE; (**b**) IgA epitope; (**c**) IgE epitope; and (**d**) IgG epitope.

**Figure 4 viruses-13-01531-f004:**
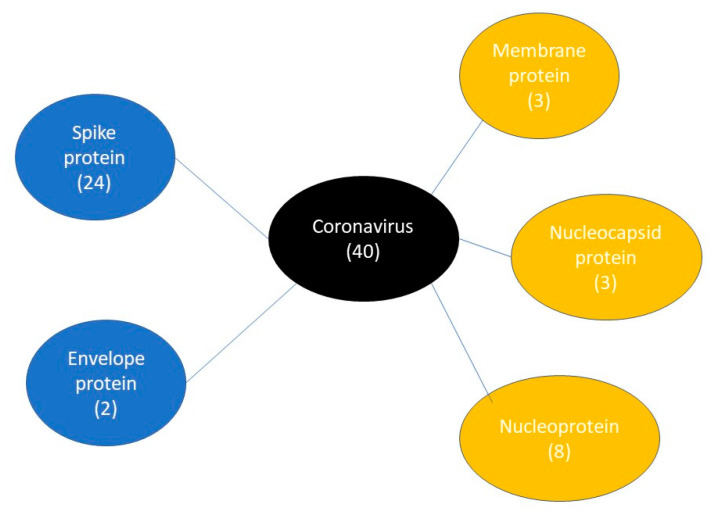
IgG epitopes related to SARS-CoV-2 protein sub-class types.

**Figure 5 viruses-13-01531-f005:**
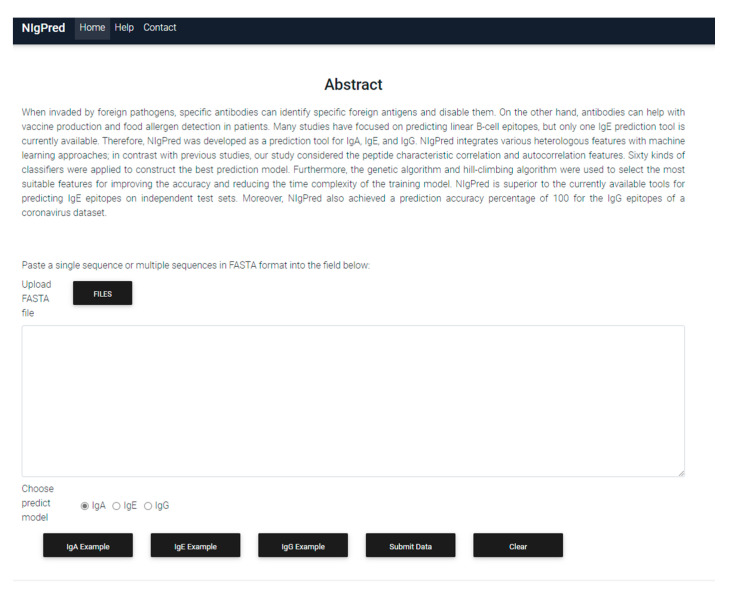
Website of the NIgPred tool.

**Table 1 viruses-13-01531-t001:** Functions of the peptides and Protr packages for feature extraction.

Feature Type	Function
Composition properties	extractAAC
	extractDC
	extractCTDD
	length
Physico-chemical properties	aindex
	boman
	mw
	crucianiProperties
	hydrophobicity
	aacomp
	blosumIndices
	PI
	mswhimScores
	kideraFactors
	fasgaiVectors
	vhseScales
	Zscales
Autocorrelation	extractMoreauBroto
	extractMoran
	extractGeary
Peptide characteristic correlation	extractSOCN
	extractQSO
	extractPAAC
	extractAPAAC
	extractCTriad

**Table 2 viruses-13-01531-t002:** Comparison of the features of different tools.

Feature Type				
Tool	Composition	Physico-Chemical	Autocorrelation	Peptide-Characteristic Correlation
IgPred	✓	✓		
BCIgEPred	✓			
NIgPred	✓	✓	✓	✓

**Table 3 viruses-13-01531-t003:** Comparison of performances using various features.

Feature Type	Features (bits)	IgA Epitope				IgE Epitope				IgG Epitope			
		Precision (%)	Recall (%)	ACC (%)	MCC (%)	Precision (%)	Recall (%)	ACC (%)	MCC (%)	Precision (%)	Recall (%)	ACC (%)	MCC (%)
Composition	526	79.5	79.5	79.5	59.0	79.0	79.0	79.0	58.0	82.9	82.5	82.5	65.4
Physico-chemical	59	80.3	80.3	80.3	60.6	73.4	73.1	73.1	46.5	81.3	81.1	81.0	62.4
Autocorrelation	96	72.9	72.9	72.8	45.7	67.0	67.0	66.9	33.9	69.5	69.5	69.4	38.9
Peptide characteristic correlation	451	85.2	85.0	85.0	70.2	79.9	79.9	79.9	59.9	84.7	84.6	84.6	69.4
All	1132	85.4	85.3	85.3	70.7	83.9	83.9	83.9	67.9	98.2	98.2	98.1	91.1

**Table 4 viruses-13-01531-t004:** Test results for the IgE, IgG, and IgA models.

Model	Model	Precision (%)	Recall (%)	ACC (%)	MCC (%)
IgA	Training	85.4	85.3	85.3	70.7
	Testing	85.8	85.8	85.7	71.5
IgE	Training	83.9	83.9	83.9	67.9
	Testing	82.0	82.0	81.9	64.0
	BCIgEPred	68.6	31.6	56.9	18.7
IgG	Training	98.2	98.2	98.1	91.1
	Testing	87.2	87.2	87.2	74.5

**Table 5 viruses-13-01531-t005:** Test results of the IgA, IgE, and IgG models using data from previous years.

Model	Train (Year)	Precision (%)	Recall (%)	ACC (%)	MCC (%)	Test (Year)	Precision (%)	Recall (%)	ACC (%)	MCC (%)
IgA	1985–2017	89.5	89.5	89.4	79.0	2018–2020	76.9	75.1	75.1	47.4
IgE	1976–2015	81.1	80.8	80.8	61.5	2015–2020	89.7	89.2	89.2	76.8
IgG	1970–2017	89.4	89.0	88.9	76.8	2017–2020	88.1	84.3	84.2	52.2

**Table 6 viruses-13-01531-t006:** Different database accuracies of IgE models when comparing tools.

Database\Tool	IgPred	BCIgEPred	NIgPred
ADFS	0.10	0.52	0.84
Allerbase	0.46	0.25	0.74

**Table 7 viruses-13-01531-t007:** List of SARS-CoV-2-related epitopes based on the NIgPred prediction results.

Antibody ID ^a^	PMID	Protein Sequence	Sub-Class Type	Actual	IgG Prediction
AB_2833115	27627203	TDVSTAIHADQLTPAWRIYSTG	Spike protein	IgG	IgG
AB_2833108	2464703	LLGCIGSTC	Spike protein	IgG	IgG
AB_2833113	16731915	GVLTPSSKRFQPFQQFG	Spike protein	IgG	IgG
AB_2833123	1711257	MKRSGYGQPIA	Spike protein	IgG	IgG
AB_2833135	16378996	ISGINASVVNIQKEIDRLNEVAKNLNESLIDLQELGKYEQYI	Spike protein	IgG	IgG
AB_2833139	16378996	FFSPQIITTDNTFVSGNCDVVIGIINNTVYDPLQPELDSF	Spike protein	IgG	IgG
AB_2833187	15184071	PDPLKPTKR	Spike protein	IgG	IgG
AB_2833188	15184071	KLRPFERDI	Spike protein	IgG	IgG
AB_2833192	20168090	FSPDGKPCTPPALNCYW	Spike protein	IgG	IgG
AB_2833197	1281870	TTGYRFTNFEPFTV	Spike protein	IgG	IgG
AB_2833217	16725238	SVYAWERKKISNCVADY	Spike protein	IgG	IgG
AB_2833224	19951177	SNVYADSFVVKGDDVRQIAP	Spike protein	IgG	IgG
AB_2833227	15194798	FQQFGRDVSDFTDSVRDPKT	Spike protein	IgG	IgG
AB_2847999	30355437	TKPLKYSYINKCSRLLSDDRTEVPQ	Spike protein	IgG	IgG
AB_2847979	16132115	SPDVDLGDASGINAS	Spike protein	IgG	IgG
AB_2847997	19951177	VYAWERKKISNCVADYSVLYNSTF	Spike protein	IgG	IgG
AB_2833160	16920216	DMDDFSRQLQ	Nucleoprotein	IgG	IgG
AB_2833161	23123213	DLIARAAKI	Nucleoprotein	IgG	IgG
AB_2833162	16920216	KKKKTDEAQP	Nucleoprotein	IgG	IgG
AB_2833173	23123213	FGPRTK	Nucleoprotein	IgG	IgG
AB_2833179	16920216	GNSRNSTPGS	Nucleoprotein	IgG	IgG
AB_2833222	15528730	KKQPTVTLLPAADMDDF	Nucleoprotein	IgG	IgG
AB_2833126	27973413	YSTEART	Membrane protein	IgG	IgG
AB_2848022	19851732	FATFVYAK	Membrane protein	IgG	IgG
AB_2833158	23025700	WAFYVR	Membrane protein	IgG	IgG
AB_2848062	31518629	DAPVFTPAP	Nucleocapsid protein	IgG	IgG
AB_2848076	7516595	PDMAEEIAALVLAKLGKDA	Nucleocapsid protein	IgG	IgG
AB_2848083	7516595	SHEAIPTRFAPGTVLPQGFYVEGSGR	Nucleocapsid protein	IgG	IgG
AB_2833199	7516595	AGQPKQVTKQSAKEVRQKILNKPRQKRTP	Nucleoprotein	IgG	IgG
AB_2833150	30877355	VAAVKDALKSLGI	Nucleoprotein	IgG	IgG
AB_2847971	1693663	SSFSSYGEI	Spike protein	IgG	IgG
AB_2847978	26292945	HDFVADMYQLAQ	Spike protein	IgG	IgG
AB_2847978	26292945	LMQINPTYYQIM	Spike protein	IgG	IgG
AB_2847978	26292945	MQYVYTPTYYML	Spike protein	IgG	IgG
AB_2847978	26292945	WSFNPSTYTIAG	Spike protein	IgG	IgG
AB_2848046	18400422	YSNIGVCK	Spike protein	IgG	IgG
AB_2848064	18400422	LQDGQVKI	Spike protein	IgG	IgG
AB_2848109	28647506	PVLVYSNIGVCKS	Spike protein	IgG	IgG
AB_2848088	9874660	MADPNRFRG	Envelope protein	IgG	IgG
AB_2848090	9874660	DPEDSALL	Envelope protein	IgG	IgG

^a^ Antibody ID is the research resource identifiers (RRID) of COVID-19 dataset from The Antibody Registry.

## Data Availability

Data is contained within the article and Appendix A.

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
