# Peer review of "NIgPred: Class-Specific Antibody Prediction for Linear B-Cell Epitopes Based on Heterogeneous Features and Machine-Learning Approaches"

_viruses, 2021, doi:10.3390/v13081531_

Round 1
Reviewer 1 Report
The manuscript describes a new tool to predict epitopes specific to immunoglobulin subtypes. It follows the standard of the field to process the input data, feature extractions, model building, and evaluations.
There are few points that authors can improvise the manuscript.
- The model could easily be over-optimized and the authors have only 5-fold cross-validation to rule this out. Their performance on the internal seems to vary a lot. For example, MCC values for IgA predictions are 66 on training, 72 on testing, and 33 on another testing set. This kind of variation raises questions of over-optimization.
- The variance among different types of datasets was shown to really make a point that only machine learning methods are required. If the variance is low between validation and training set, the higher performance points to the over-optimized models.
- Further looking at the comparison with existing tools, raises eyebrows. Authors should offer some explanation, how this high performance is achieved with their model, which was missing previously.
- The complete picture with the number of epitopes and antigens should be added.
- The authors should also shed some thoughts on discontinuous epitope predictions. At least mention these limitations.
- The authors should also check the citations, like citing the latest publication of IEDB database, not the old versions. The same is true for any other tool/database used.
- CD hit clustering largely depends upon the order of sequences provided in the input. Any thoughts on this?
Reviewer 2 Report
This manuscript describes a new tool for predicting linear epitopes that are isotype-specific. The performance of the underlying algorithm is shown to improve over other available tools. While on face value, the reported results appear to be strong, there are a number of areas where this study can be improved upon. In particular:
- An underlying implication of this study is that there are fundamental differences in epitope specificity for different isotypes. However, what is the biological evidence for this? For example, there can be IgG and IgE variants of the same B cell receptor that can result from class-switching and antigen affinity maturation, but we would not expect the epitopes for these IgG and IgE relatives to be different. In fact, in general, the epitope specificity of a B cell is determined from the Fab region (and more precisely – from FWR1-FWR4 regions of both the heavy and light chains). What this study suggests appears to be counterintuitive, and will benefit from additional analysis that may help support the underlying assumption about differences between isotypes. For example, do IgA antibodies appear to be more closely similar to each other in their Fv regions, compared to IgG or IgE antibodies?
- It will be helpful to include a figure that highlights what is new in NIgPred compared to previous methods. The new algorithm appears to be substantially better than the competitors, but it is not immediately clear what mechanisms may explain this gain in performance. There is some discussion of this, but it will be helpful to better visualize this in some way.
- It will be informative to do more analysis on the prediction performance of the algorithm. Are there any specific sequence features that are associated with greater prediction accuracy? Does prediction accuracy correlate with how closely an antibody from the test set matches antibody sequences from the training set?
- The statements on lines 190-193 do not seem to match the data in Figure 2. It is also not clear exactly what Figure 2 shows. Can statistics be added to help quantify differences?
- Figure 1 – typo: one should say ‘IgG’; there are two ‘IgE’ currently.
- Line 236 – should say IgA, not IgE.
- Language – many typos throughout; review for grammar as well.
Round 2
Reviewer 2 Report
I appreciate the revisions that the authors made to the original version of the manuscript. Most of the comments were addressed, but my major concern is that I still do not understand the fundamental reasons for doing separate analysis for different isotypes. This implies that there are fundamental differences between the epitopes targeted by isotypes, but it is not clear what these differences are - otherwise, why not do analysis for a combined dataset? There also seem to be some remaining language issues in a few places in the manuscript.
Author Response
I appreciate the revisions that the authors made to the original version of the manuscript. Most of the comments were addressed, but my major concern is that I still do not understand the fundamental reasons for doing separate analysis for different isotypes. This implies that there are fundamental differences between the epitopes targeted by isotypes, but it is not clear what these differences are - otherwise, why not do analysis for a combined dataset? There also seem to be some remaining language issues in a few places in the manuscript.
Response:
Thanks for your advice. We tried to compare the datasets of IgA, IgE, and IgG concerning whether the composition of 20 amino acids is different. We used a one-way ANOVA test with a P-value greater than 0.05, indicating that there was no significant difference in the amino acid composition of the three datasets. However, we observed the important features that the feature selection algorithm ordered after training for the three models, and found that there were significant differences in the important features among them. Therefore, we reorganized Supplementary Table S1 and sorted the more important features of the three models first after feature selection.
The results showed that the composition properties are the most important features in the IgA model, followed by physicochemical properties and peptide characteristic correlation. In the IgE model, more attention is paid to composition and physicochemical properties, and the detailed feature functions of composition properties are completely different from that of the IgA model. In the IgG model, the top-1 significant feature is the length of the epitope, which is the same as in the IgA model, but most others use peptide characteristic correlation. So what we can conclude from this is that the three models have their unique features/characteristics, and it's not appropriate to mix them up. Three different models should be tested separately to predict whether a particular epitope sequence would be recognized by a specific antibody.
In response, we have rewritten the entire Section 3.2. Besides, we have our manuscripts checked again by MDPI English Editing Services. We appreciate the chance to make this manuscript better.

This manuscript is a resubmission of an earlier submission. The following is a list of the peer review reports and author responses from that submission.
Round 1
Reviewer 1 Report
The manuscript entitled “NIgPred: Class-specific antibody prediction for linear B-cell epitopes by heterogeneous features and machine learning approaches” by Tung et al describes their webtool in predicting class-specific antibodies (e.g., IgA, IgE, IgG) of a given linear epitope sequence.
This paper is perhaps too technical in nature for a journal like Viruses. Additionally, it was difficult to review and it does not provide a number of details of the approach and parameters used (e.g., diamino acid composition, protein distance etc.). It may be that they are pretty commonly used in the field of study. For readers who are not familiar with the field, it is the authors responsibility to provide a brief introduction to significance of these parameters.
Fig.2: It is better to express the results as amino acid propensities rather than percentages. Even then, I am not sure how useful it is, as one cannot be very certain based on the single amino acid level, as their relative occurrences are mostly the same across the board. Authors should do this analysis for various peptide lengths.
Last but not the least, the website does not work as advertised.
If I take one of their SARS COV-2 epitopes (e.g., LLGCIGSTC) and plug it into their server
The following results are displayed. Not sure why the original sequence was not output. The Table 5 in the paper shows a score of 0.855 for this peptide. There appears to be a disconnect between the results reported and what can be obtained from the website.
Also the CLEAR button does not work.
All things considered, this paper may not be suitable and too technical in nature for publication in the journal Viruses.

Reviewer 2 Report
The manuscript "NIgPred: Class-specific antibody prediction..." by Tung et al describes a computer algorithm that they claim predicts 'linear' epitopes of IgA, IgE and IgG antibody isotypes. The manuscript is confusing in several places but seems to conclude that IgE epitopes can be better predicted based on considering the "protein distance and autocorrelation features". This author is an immunologist, not a bioinformatics expert, so has no reason to doubt that improvements were made to the prediction algorithms. However, there are many inaccurate descriptions and statements throughout the manuscript regarding the immunology and usage of terms that must be clarified, as currently the biological interpretations and implications are difficult to understand. The problematic statements are listed below. In addition, could the authors explain more clearly for the COVID-19 experiment, what is being predicted? All of the antibodies are IgG, but isn't the point of their program to discriminate between IgGs and IgA/IgE? Why wasn't this tested and isn't that more relevant?
Some specific comments:
Line 48- (2nd para) – This second paragraph has several inaccurate or erroneous statements. The statement, “Epitopes can identify foreign antigens and cause antigens to lose their ability to function, so they can help with vaccine production” is bizarre. A specific antibody is what defines a specific epitope on a particular antigen, not the other way around. What is meant by antigens losing function? Do they mean antibodies can neutralize virus, opsonize bacteria, inactivate a toxin? Is a vaccine an antigen, and if so, do antibodies make vaccines lose function? Next: “B cell epitopes can be divided into 5 categories… IgG, IgA…” No! What are listed are isotypes of B cells/antibodies, not epitopes. “With the outbreak… every scientist is looking for a way to develop a COVID-19 vaccine” Is this proper science writing?
The authors must be precise with their use of terms, avoid erroneous platitudes, etc. The manuscript should be proofread by a professional science writer.
The subsequent transition from vaccines to food allergies is also sort of confusing.
Line 61 – “…90% of epitope are structural…” This is an arbitrary statement, as all epitopes are “structural”. No reference is given either. This point must be clarified and referenced.
Line 69- “some tools…can predict the exact type of epitope” is unclear and should be worded to be more precise. Surely no prediction tool will predict an exact epitope in every case: how can a prediction algorithm discriminate an epitope based on isotype? If a BCR class-switches from IgG to another isotype, without acquiring new SHM, the variable regions of the antibody are identical by definition and a computer, therefore, cannot discriminate between them. If certain variable sequences are enriched in a particular isotype, the authors should introduce this more clearly, discuss why this may be the case, and what has been published on it in terms of the underlying mechanisms and/or assumptions involved.
Line 78- “AlgPred is a tool…” Authors should cite this tool with a reference.
Line 98- “…our model… was superior to that of the existing BCIgEPred tool”. Models can be developed and trained on existing data to the point of perfection, but the real test of a model is its ability to predict from a novel dataset to which it has not been trained. Can the authors please address this here?
Line 148 – “Monoamino acid and diamino acid compositions are the easiest and most useful features…” What is the reference or basis for this claim?
Line 151 – The authors list and cite three approaches to study peptide sequences. Can they please explain why each approach was selected?
Line 200 – “…protein distance was the most important feature for the IgE model… the length of IgE being mostly between 12 and 20” Authors must clarify what is meant by “protein distance” here, specifically? Authors should also clarify the length of what is between 12 and 20, CDRs? Fc region? Hinge region? Linear epitopes? And why would the latter explain the former?
Line 207-208 – “…IgE data are mostly of length 13-15… 5-12, 5-20, and 5-12…” What is meant by IgE data? Length of what, amino acid residues? Also, couldn’t the IgE data set be less diverse due to sampling bias and not to any particular biological explanation?